# Impact of Long Working Hours on Mental Health: Evidence from China

**DOI:** 10.3390/ijerph20021641

**Published:** 2023-01-16

**Authors:** Xinxin Ma

**Affiliations:** Faculty of Economics, Hosei University, 4342 Machita-shi Aiharamachi, Tokyo 194-0298, Japan; xxma@hosei.ac.jp

**Keywords:** long working hours, mental health, risk of mental illness, China

## Abstract

Although previous studies have examined the impact of long working hours on mental health in China, they have not addressed the initial value and reverse causality issues. To bridge this gap in the literature, I conducted a dynamic longitudinal analysis to investigate the association between long working hours and the risk of mental illness nationwide. Using three-wave longitudinal data from the China Family Panel Studies conducted in 2014, 2016, and 2018, I adopted dynamic regression models with lagged long working hours variables to examine their association with the risk of mental illness. The results indicate that long working hours have positive and significant (*p* < 0.01 or *p* < 0.05) associations with the risk of mental illness (OR: 1.12~1.22). The effect is more significant for women, white-collar workers, and employees in micro-firms, compared with their counterparts (i.e., men, pink- and blue-collar workers, employees of large firms, and self-employed individuals). The results provide empirical evidence of the effects of long working hours on mental health in China, confirming the need to enforce the regulations regarding standard working hours and monitor regulatory compliance by companies, as these factors are expected to improve mental health.

## 1. Introduction

Mental illness (MI) and long working hours are two primary issues that plague employees worldwide [1,2,3]. In 2017, 792 million people worldwide were reported to be living with an MI; they constituted 10.7% of the global population, a figure slightly higher than 1 in 10 people [1]. As medical care expenses for MI are high [1,3] and MI may reduce labor productivity, negatively affecting human capital accumulation in most countries, exploring the determinants of MI is a critical issue in the public health field. 

Some empirical studies revealed that individual attributes (e.g., education, age, sex) [4,5,6,7], social capital [8,9], and life events (e.g., marriage, fertility) [10,11,12] may affect the risk of developing MI. Furthermore, work-life conflict may increase the risk of MI [13,14]. Among the various work environment factors that may harm workers’ mental health status, prior empirical works identify long working hours as a risk factor [4,5,6,7,15,16,17,18,19,20,21,22,23]. However, most studies concentrated only on developed countries, and evidence from China is scarce.

This study focuses on the relationship between long working hours and MI in China. China was selected as an analyzed target because it is the most populated country in the world and has the highest number of patients with MI, globally. According to the World Health Organization, approximately 54 and 41 million people in China suffer from depression and anxiety disorders, respectively; the proportion of Chinese people with MI is more than 12% of that worldwide [3]. Hence, a public health policy aimed at improving Chinese mental health status may contribute to population welfare worldwide. In addition, regarding overtime work in China, although the Labor Law of the People’s Republic of China regulates the standard working hours (around 40 h weekly), work hours are actually longer in China’s privately owned enterprises and the informal sector (e.g., the self-employed) [24,25]. Although long working hours likely affect a worker’s mental health in China, empirical studies on the issue that focus specifically on China are scarce. This study attempts to fill the above-mentioned gaps. 

Regarding the relationship between long working hours and MI, from the occupa- tional health and human resource management theory perspectives, two models were advocated to indicate that long working hours negatively affect mental health: first, according to the job demand-control model [26], involuntary long working hours may lead to MI because of the imbalance between work responsibility (the reality of long working hours) and authority (wherein employees lack authority to determine their own working hours); and second, according to the effort-reward imbalance model [27], when the efforts involved in involuntary long working hours are not rewarded (e.g., through unpaid overtime or low overtime premium), the probability of MI may become higher among those working long hours. From the labor and family economics theory perspectives, based on the individual/household utility model, an individual will maximize the utility based on income and time constraints. An individual’s hours can be divided into three parts: market work, housework, and leisure [28,29], indicating a trade-off relationship between working hours and housework/leisure hours. Therefore, long working hours may reduce housework/leisure hours and generate work-life conflict (especially for female workers during their motherhood period), which may decline an individual’s utility and lead to MI [30,31,32]. 

Numerous studies have emphasized that long working hours are a primary contributor to MI for the working-age population e.g., [4,5,6,7,15,16,17,18,19,20,21,22,23], in addition to demography [4,5,6,7], family factors [10,11,12], social factors, such as social participation, and social capital [8,9]. These studies focus on developed countries and have reported that long working hours negatively affect mental health e.g., [4,5,6,7,15,16,17,18,19,20,21,22,23]; however, equivalent studies for China are scarce [33,34]. This study empirically investigated the association between long working hours and mental health in the context of China, and makes three contributions to the related literature. First, while two empirical studies have focused on the issue, they used the cross-sectional analysis method, raising the possibility of estimation bias in the results. By contrast, this study was based on three-wave longitudinal survey data from the China Family Panel Studies (CFPS) and addressed statistical issues, such as the initial value effect (i.e., the effect of a variable’s initial value on its current value) and reverse causality—problems that have remained largely unsolved in previous studies [33,34]. Second, unlike previous studies that concentrated on one definition of “long working hours,” this study used a different definition (e.g., the different cut-off values to define long working hours) to perform robustness checks. Third, although some studies for developed countries have reported that the effect of long working hours differs by sex, educational attainment, occupation, and age group [15,16,17,18,19,20,21,22,23], no study has analyzed these differences specifically for China, and this study is the first. 

According to the job demand-control [26] and effort-reward imbalance models [27], and the previous studies above, I hypothesize that long working hours may negatively affect Chinese workers’ mental health status, and the negative effect will differ by group. I perform an empirical study to prove our hypotheses in the following.

## 2. Methods

### 2.1. Data

This study used the latest three-wave data obtained from the CFPS, a nationwide longitudinal survey conducted by Peking University in representative regions of China in 2014, 2016, and 2018. The CFPS sample is a multi-stage probability sample drawn using the implicit stratification method. In the selected sample of villages/residential committees in provinces and municipals, the household list was obtained from the survey map for the end sampling frame and used following the cyclic equidistant sampling method with random starting points to expand the sample size and extract household samples. The survey interviewees included all family members in a household living in the same region for more than six months. 

The reasons for using the CFPS are as follows. First, the CFPS is a representative nationwide survey that has been used in numerous empirical studies on China e.g., [35,36,37]. It collects individual, family, and community-level longitudinal data in contemporary China. The sample for the 2010 CFPS baseline survey was drawn through multistage probability with implicit stratification. In the 2010 baseline survey, the CFPS successfully interviewed approximately 15,000 families and 30,000 individuals within these families, with an approximate response rate of 79%. The respondents were tracked via follow-up surveys. The sample sizes of the CFPS data for 2014, 2016, and 2018 were 37,147, 36,892, and 37,354, respectively. The CFPS covered 25 provinces and municipalities in 2010 and 31 provinces in the current survey. I declined the special regions, such as Tibet, Xingjian, and Inner Mongolia because their samples were less than 10. Second, the CFPS contains rich individual- and household-level information that can be used in the empirical study, such as a set of indices on mental health status, working hours, demographic characteristics, and the number of family members, workplace, and region. 

The samples include Chinese working individuals in the urban and rural areas and in the formal and informal sectors, including the employees and self-employed individuals. According to the Labor Law of the People’s Republic of China, the legal working age is 16 years. Based on the employee basic pension insurance policy, the mandatory retirement age is 60 years for male workers and cadres, 55 years for female cadres, and 50 years for female workers; therefore, I selected the samples aged 16–60 years in the baseline survey who had committed to at least one of two follow-up surveys. After excluding respondents with missing data on the key variables used in the statistical analysis, the data for 21,093 individuals were applied to this study (6,328 in 2014; 6,117 in 2016; and 8,648 in 2018), which is an unbalanced panel dataset. I weighted the samples using the nationwide population size in this study.

### 2.2. Variable Settings

I conducted the dependent variable, key independent variable, and confounding variables as follows. The key dependent variable was a binary variable of risk of MI, which was constructed based on the following six types of mental health status that were common question items in the CFPS from 2014 to 2018: (i) I find nothing exciting; (ii) I feel nervous; (iii) I cannot concentrate on things; (iv) I feel depressed; (v) I find it difficult to do anything; and (vi) I feel that I cannot continue with my life. The answer options to the question “How often do you think about your mental health status?” for each question were: 5–7 days weekly, 3–4 days weekly, 1–2 days weekly, less than 1 day weekly, or never. We coded the value of the category options as “5–7 days weekly = 4; 3–4 days weekly = 3; 1–2 days weekly = 2; less than 1 day weekly or never = 1,” and calculated the total score of MI, ranging from 6–24. Mental health was part of the original CFPS questionnaire and was used for the first time in this study; a high value indicated a higher risk of MI (higher likelihood of developing mental illness). I constructed a binary variable, MI, which equals 1 when the total mental health score ≥ 10 and 0 when the score < 10. I also used the three cut-off values, 12, 16, and 20, for the robustness checks. The results were consistent, and I reported them using the lowest value (10) as the cut-off value to construct the main indicator of the risk of MI in this study.

The key independent variable was the long working hours dummy variable. Referring to previous studies [4,5,6,7,15,16,17,18,19,20,21,22,23], based on the question, “How many hours did you spend at work each week?,” I constructed the long working hours dummy variable with a value of 1 for weekly working hours ≥ 50 and 0 for weekly working hours < 50 for basic analysis. I also used two other definitions of long working hours variables to implement the robustness checks: (i) to change the cut-off value of long working hours as 1 = weekly working hours ≥ 60 and 0 = weekly working hours < 60; (ii) a set of dummy variables of working hours: weekly working hours < 35; weekly working hours ≥ 35 and < 39; weekly working hours ≥ 40 and <50; weekly working hours ≥ 50 and<60; and weekly working hours ≥ 60.

Based on previous studies [4,5,6,7,15,16,17,18,19,20,21,22,23,38,39], I considered the following confounding variables, all of which were likely to affect mental health status and were available from the CFPS: (1) socio-demographic factors, including age, sex, years of education, ethnicity (Han), Communist Party of China membership, urban residence; (2) family factors, including the presence of a spouse and number of family members; (3) large company size: number of employees ≥ 500; (4) occupation (manager and technician, staff and service, operation worker, and others); (5) industry sector (1 = manufacturing industry, 0 = other industry sector); (6) enrollment pension/medical insurance (1 = enrollment, 0 = otherwise); (7) region (east, central, and west); and (8) survey years (2014, 2016, and 2018). 

### 2.3. Analytic Strategy

As the benchmark, I considered the logistic regression model to estimate the association between long working hours and MI, along with a set of covariates, *X*:(1)$$MI_{i}=a+\beta LWH_{i}+\sum_{n}\delta_{n}X_{ni}+\varepsilon_{i}$$
where MI denotes the risk of MI; *i* and *n* denote the individual and number of covariates, respectively; LWH denotes the indicator of long working hours; X denotes the covariates; β and δ are the coefficients of LWH and X*,* respectively, a is a constant term; and *ε* is an error term. 

I addressed the initial value problem [40,41]: mental health status at time *t* may be affected by mental health status at time *t−n*. To resolve this problem, I considered a dynamic model that included the risk of MI at time *t−n* as an explanatory variable. I further addressed the reverse causality issue using overtime work status at time *t−n* to mitigate the problem by allowing a one- or two-wave lag between long working hours and mental health [42,43]. Overall, I estimated the following dynamic lagged variable (LV) model:(2)$$MI_{it}=a+\rho MI_{it-1}+\beta LWH_{it-n}+\sum_{n}\delta_{n}X_{nit}+u_{it}$$
where *t* and *t* − *n* denote combinations of survey year and prior year, respectively. As I used the three waves (2014, 2016, 2018) of longitudinal data, we used the combinations of 2014 and 2016, 2016, and 2018 in the LVt-1 model and used the combination of 2014 and 2018 in the LVt-2 model. 

I estimated these models using subsamples by sex (men and women), education (lower than university, university, and higher), age (aged 16–29, 30–44, 45–60), and occupation to compare heterogeneous groups. Referring to existing studies [30,31], I divided the sample into three occupational groups: white-collar (manager or technician), pink-collar (service or staff), and blue-collar (operation worker). 

## 3. Results

### 3.1. Descriptive Analysis

To compare the differences in demographic or social-economic status factors between groups during the analyzed period, I used the pooling data of three waves and weighted them by nationwide population size. Table 1 summarizes the key features of the sample used for statistical analysis. I calculated the mean values by three groups: total, long working hours (working hours ≥ 50), and standard or short working hours (working hours < 50) and weighted them by population size. The *p*-value in the *t*-test indicates the statistical significance of the differences in mean values of these variables between long- and standard or short-working-hours groups. 

First, based on the results in column (a), in general, 46% of the respondents experienced long work hours from 2014 to 2018. The proportion of the high-risk MI group (whose MI score was higher than 12) is 32%. The average years of schooling of the group aged 16–60 was 8.13 years; the proportion of urban residents was 45%, and the distribution of the Han majority was high at 95%. Moreover, 17% of workers were in the manufacturing industry sector, and 61% and 92% of individuals participated in public pension and medical insurance. 

Second, the demographic, family, and workplace factors differ in the long- and standard or less working hours groups. For example, the proportion of women was 49% for the total sample, 42% for the long working hours group, and 55% for the standard or less working hours group, suggesting a gender distribution gap in the two groups. The proportion of operation workers was 26% for the total sample, 35% for the long working hours group, and 19% for the standard or short working hours group, indicating that the occupational distribution differs among different working hour groups. 

Third, the proportion of the high-risk MI group (whose MI score is higher than 12) was slightly higher for the long working hours group (33%) than that of their counterpart (31%). Although the difference (2%) between the two groups is not large, the *t*-test results indicate a statistically significant difference, possibly because of the large sample size.

However, these results did not control the confounder factors listed above that may significantly affect the risk of MI. 

### 3.2. Regression Analysis

The results of the three logistic regression models (general logistic, LVt-1, and LVt-2) are summarized in Table 2, which reports the odds ratios (ORs) of MI, along with the 95% confidence intervals (CIs), in response to long working hours, after controlling for all covariates. The table shows that long working hours have positive and significant (*p* < 0.01 or *p* < 0.05) associations with the risk of MI ([Model1] OR: 1.22, 95% CI: 1.11,1.33; [Model 2] OR: 1.12, 95% CI: 1.01,1.24; [Model 3] OR: 1.12, 95% CI: 1.01,1.24). Furthermore, the coefficients of the initial independent variables (MIt-1) are significant at 1% levels, suggesting an initial problem. Hence, using the dynamic model is valid.

Table 3 displays the results of the robustness checks. First, I changed the definition of long working hours to “weekly working hours ≥ 60;” the results show that long working hours had positive and significant (*p* < 0.01, *p* < 0.05, or *p* < 0.1) associations with the risk of MI ([Model1] OR: 1.23, 95% CI: 1.12,1.36; [Model 2] OR: 1.11, 95% CI: 0.98,1.25; [Model 3] OR: 1.16, 95% CI: 1.03,1.30). 

Second, I replaced the binary variable of long working hours with a set of dummy variables; the results indicate that compared with the short working hours group (working hours < 35), the likelihood of becoming MI was significantly (*p* < 0.01, *p* < 0.05 or *p* < 0.1) higher for the long working hours group (working hours ≥ 60) ([Model1] OR: 1.22, 95% CI: 1.08,1.37; [Model 2] OR: 1.03, 95% CI: 0.87,1.17; [Model 3] OR: 1.23, 95% CI: 1.07,1.42). 

Third, I replaced the binary variable for the risk of MI using different cut-off values. I used three higher values (12, 16, and 20) for the total mental health score to create three new indicators of the risk of MI. I re-ran the estimations. The results are similar to those in Table 2. For example, when using the higher cut-off value of 20, the magnitudes of the ORs ([Model1] OR: 1.21, 95% CI: 1.11,1.32; [Model 2] OR: 1.15, 95% CI: 1.03,1.28; [Model 3] OR: 1.13, 95% CI: 1.02,1.25) are slightly larger in the LV models than in Table 2, and the results verify the conclusions. 

Fourth, considering that the majority of laborers in the agricultural industry sector were self-employed and can adjust working hours independently, thereby reducing the negative effect of long working hours, I estimated these models, excluding farmers. The findings ([Model1] OR: 1.39, 95% CI: 1.20,1.60; [Model 2] OR: 1.17, 95% CI: 0.98,1.39; [Model 3] OR: 1.13, 95% CI: 0.96,1.34) are also similar to those in Table 2. These results confirmed the positive association between long working hours and the risk of MI in China. 

Finally, I used the logit regression model to estimate the coefficients instead of the OR, which are positive values and statistically significant at 1% or 5% levels, thus confirming the conclusions. 

Table 4, Table 5 and Table 6 summarize the results obtained from separate estimations by sex; occupation; and employment sector group. The main results are as follows. First, although the coefficients of long working hours are positive values for both men and women, they are not statistically significant for either. The positive effect of long working hours on the risk of MI is slightly greater for women than for men (1.13; *p* < 0.1 for women). 

Second, the positive effect of long working hours on the risk of MI is significant only in the white-collar group (OR: 1.65, *p* < 0.05). 

Finally, based on the questionnaire items of occupation and the size (number of workers) of the workplace, and the definitions of the individual (small business, *geti qiye*) and large size firms by the National Statistics Bureau in China, we distinguished three types of employment sector: (1) the self-employed individuals, including the employers of small businesses, and own-account workers; (2) employees in a micro-firm with the number of workers less than 7; and (3) employees in a large firm with the number of workers more than 500. (1) and (2) can be considered the informal sectors, whereas (3) can be considered the formal sector. The positive effect of long working hours on the risk of MI is only significant for the employees in a micro-firm (OR: 1.29, *p* < 0.05). 

## 4. Discussion

This study examined how long working hours were associated with the risk of MI in China for the period of 2014 to 2018. The regression analysis based on three-wave longitudinal data and dynamic models, with the lagged working hours variable indicating that long working hours had a significant positive association with the risk of MI. The basic results are consistent with the findings of previous cross-sectional studies in China [33,34], which did not fully control for statistical biases; however, the OR values are higher in this study. These findings contribute to the literature on the association between long working hours and mental health, while providing richer and more robust conclusions. These estimation results shed light on the association between long working hours and mental health in depth as follows.

First, the estimation results in this study confirmed that long working hours may increase the probability of having an MI. The magnitude of the OR ratio coefficients (1.12–1.22 in Table 3) was smaller than those in other studies for China (1.745) [15]. These differences may be attributable to the different data and models: we used the three-wave national longitudinal survey data from 2014 to 2018 and the dynamic lagged variable model to address the initial dependent and reverse causality issues, whereas previous studies used one-point cross-sectional survey data and did not address the bias in estimations. To compare the effects of long working hours on the risk of MI between China and other countries, according to a meta-analysis [15], the effect was smaller for China than for other Asian countries, such as Japan (1.333) and Korea (1.237), and other Western countries, such as Italy (1.341), Spain (1.248), and the United States (1.274), but greater than that for Denmark (1.091), Finland (1.063), and the United Kingdom (1.083). The international comparison results indicated that the institutional and cultural differences among countries may affect the effects of long working hours on mental health, which should be analyzed in detail in the future.

Second, the results indicated that the negative effect of long working hours on mental health was modestly more significant for women than for men; this may be because of the division of gender roles in familial responsibility. Although the Chinese government has promoted employment equality in the labor market, women typically bear higher family responsibility (e.g., childcare, parent care, and housework) than men [44,45]; the double shift (long working hours followed by long housework hours) may lead to more family-work conflict for women than men, thereby enhancing the negative effect of long working hours for women. 

Third, a disparity existed in the effects of long working hours among various occupational groups. The results indicated that the negative effect of long working hours on mental health was greater for white-collar workers than their counterparts (pink- and blue-collar groups). This may be because of the differences in job content between these groups: white-collar workers may face more challenges in terms of addressing changes in the environment or fostering innovation, thereby increasing their work stress; by contrast, their counterparts primarily work routine jobs [15,16,46,47]. Based on the effort-reward imbalance model [27] and individual/household utility model [8,9], when the efforts exceed the reward, the trade-off of working and leisure hours may generate more work-family conflict and increase the risk of MI among white-collar worker groups than their counterparts.

Fourth, the negative effect of long working hours on the risk of MI was more significant for the employees in the informal sector (micro-firms) than those in the formal sector (large-sized firms). Labor Law and social security policies are not implemented in the informal sector [48], and there are almost no trade unions in the informal sector; therefore, the employees in this sector constitute a disadvantaged group that might be forced to work longer hours. Based on the job demand-control model [26], because the employees in the informal sector cannot control the working hours to address the high job demand, the risk of MI will be greater. Notably, we found that the negative effect of long working hours on mental health was insignificant for the self-emplyed individual group in the informal sector. This might be because self-emplyed individuals can control and adjust the working hours on their own, thereby reducing the adverse effects. Moreover, the self-emplyed individual group might choose to work long hours voluntarily, which may also reduce the negative effect of long working hours on mental health. 

Based on the results of this study, we can argue that, in general, policies to reduce long working hours may improve the Chinese population’s mental health status. First, although the Labor Law of the People’s Republic of China mandates that the standard work hours be less than 40 hours weekly, the proportion of workers who worked for 50 hours or more was 46% based on the CFPS data from 2014 to 2018, suggesting that a compliance problem remains in most Chinese companies. Working hours were reported to be longer in China, especially for privately-owned companies [24,25]; therefore, policy suggestions include: (1) monitoring the compliance of regulatory policies regarding working hours in companies (especially privately owned); (2) policies should be implemented to reduce long working hours or provide more social services to reduce the housework burden (e.g., providing more public kindergartens), which may contribute to improving the mental health status; (3) unlike in most developed countries (e.g., Japan), a mental health counseling department is lacking in most Chinese companies; the policy to promote the establishment of a mental health counseling center at the workplace is expected to improve mental health status in China; and (4) as the negative effects of long working hours on mental health differ by group, the effects of these policies may also differ by group, suggesting that detailed measures are necessary for different groups. 

This study has several limitations. First, the results may be subject to measurement [49,50]. For example, women may report a higher risk of MI than men, and the less educated and older groups might not report the correct status of their mental health because of the lack of literacy, so future studies should aim to adopt objective indicators of MI. The measurement of long working hours (e.g., measurement scale, cut-off point, and the definition of long working hours) differs across empirical studies [28], and an international comparison cannot be easily performed. Thus, international comparison research based on similar definitions should be pursued in the future. Second, although the sample for the baseline of the CFPS was drawn through multistage probability with implicit stratification, it may also maintain the plots or sample selection bias in the survey. Third, although we used dynamic models with lagged long working hours variables to address the reverse causality problem, we could not identify the underlying causality of long working hours affecting mental health such as the individual heterogeneity and self-selection problems, which should be investigated in greater depth using various quantitative study models (e.g., difference-in-differences, instrumental variable methods). Fourth, as China is a developing and emerging economic country, and the influence of government (or Communist Party of China organization) on firms is greater than that in developed countries [51,52], it is important to account for international comparisons between China and other counties while considering these institutional differences. We should control these institutional and cultural factors in any international comparison. Fifth, because we could not obtain appropriate information (e.g., the job allocation, effort or willingness to work, work environment in the workplace) from the questionnaire items of the CFPS, future work could conduct alternative surveys such as employer-employee surveys that include these workplace items and employ empirical methods to explore the mechanism by which long working hours affect mental health status. Finally, the COVID-19 pandemic has immensely changed employment status [53,54], thus damaging individuals’ mental health status worldwide [55,56]. Further study on employee mental health in light of the COVID-19 pandemic based on a new survey or following-up longitudinal survey data is a new challenge. Although we analyzed data from a survey conducted before the COVID-19 pandemic, employment has slowly returned to normal in the recent period in most countries. The Chinese government began winding down its policy of closing down cities in December 2022. Due to the fact that the COVID-19 pandemic negatively affected economic growth worldwide, which led to dramatic decreases in labor demand in the past three years, employers will extend employees’ working hours to reduce labor costs. Hence, the problem of long working hours may become much more serious in the future. The results from this empirical study and the existing research conducted in developed countries also suggest that the government should enforce working hour regulations to reduce the risk of MI during or after the COVID-19 pandemic period.

## 5. Conclusions

Based on the analysis of the three-wave longitudinal data from 2014 to 2018, I conclude that long working hours are negatively associated with the mental health of workers aged 16–60 years in China, and that the negative effects of long working hours differ by sex, occupation, and employment sector groups. 

Despite its limitations, this study, which employed longitudinal data, provides novel insights for understanding the association between long working hours and mental health in China. Moreover, this study is the first to compare the effects of long working hours by sex, occupation, and employment sector groups in China. I expect the Chinese experience to provide valuable lessons for other countries looking to improve their nations’ mental health and populations’ welfare.

## Figures and Tables

**Table 1 ijerph-20-01641-t001:** Differences in mean values of variables by long and non-long working hour groups.

	(a) Total	(b) WH ≥ 50	(c) WH < 50	Difference	
				(b)–(c)	*t*-test
WH (hours)	46.15	64.95	29.83	35.12	*p* < 0.000
LWH	46%				
MI	32%	33%	31%	2%	*p* < 0.000
Demographic factors					
Education (years)	8.13	8.01	8.24	−0.23	*p* < 0.000
Age (years)	42.04	41.17	42.79	−1.62	*p* < 0.000
Women	49%	42%	55%	−13%	*p* < 0.000
Urban	45%	45%	45%	0%	*p* < 0.135
Ethnicity (Han)	96%	95%	96%	−1%	*p* < 0.051
Party membership	5%	4%	6%	−2%	*p* < 0.000
Family factors					
Married	90%	90%	90%	0%	*p* < 0.052
Number of family members	4.44	4.47	4.42	0.05	*p* < 0.000
Large company size	9%	8%	9%	−1%	*p* < 0.006
Occupation					
Manager and technician	7%	5%	9%	−4%	*p* < 0.000
Service	21%	22%	19%	3%	*p* < 0.000
Operation worker	26%	35%	19%	16%	*p* < 0.000
Others	46%	38%	52%	−14%	*p* < 0.000
Industry sector					
Manufactural	17%	20%	14%	6%	*p* < 0.000
Social insurance					
Pension	61%	60%	62%	−2%	*p* < 0.002
Medical insurance	92%	92%	93%	−1%	*p* < 0.028
Regions					
West	31%	33%	30%	3%	*p* < 0.000
Central	29%	28%	30%	−2%	*p* < 0.000
East	40%	39%	40%	−1%	*p* < 0.061
*N*	21,093	9805	11,288		

Note: WH: weekly working hours; LWH: long working hours; MI: risk of mental illness.

**Table 2 ijerph-20-01641-t002:** Estimated associations between long working hours and MI.

	(1) Logit			(2) LVt-1_Logit			(2) LVt-2_Logit		
	OR.		95% CI	OR.		95% CI	OR.		95% CI
MIt-1	2.84	**	(2.54,3.17)	2.67	**	(2.38,2.98)	2.88	**	(2.51,3.29)
LWH(WH ≥ 50)	1.22	**	(1.11,1.33)	1.12	*	(1.01,1.24)	1.12	*	(1.01,1.24)
Covariates	Yes			Yes			Yes		
*N*	11010			11010			7060		
Log likelihood	−5994.556			−4552.136			−4403.115		
Pseudo R2	0.103			0.117			0.055		

Note: Obtained the results from the dynamic logistic models with lagged explanatory variables. Logit: logistic regression model. LVt-1: model using the indicator for the risk of mental illness in the previous period. LVt-2: model using the indicator for the risk of mental illness in the past two periods. OR: odds ratio. CI: confidence interval. MIt-1: the indicator of the risk of mental illness in the prior survey year. LWH: a dummy indicator variable for long working hours, equal to 1 when an individual answered that their weekly work hours were 50 h. or more and 0 otherwise. WH: weekly working hours. The covariates including years of education, years of working experience and its squared, female, ethnicity (Han dummy), party membership, urban, married, number of family members, occupation (manager and technician, staff and service, and other occupations), industry sector (manufacture dummy), participation in public pension, participation in medical insurance, region (central, west) and year dummy variables (2014, 2016) were controlled. The reference group is operation workers, east region, and 2018 year. ** *p* < 0.01, * *p* <0.05.

**Table 3 ijerph-20-01641-t003:** Results of the robustness checks by using different definitions of long working hours.

	(1) Logit			(2) LVt_1_Logit			(2) LVt_2_Logit		
	OR.		95% CI	OR.		95% CI	OR.		95% CI
(1) Change the definition of LWH (WH ≥ 60)									
MIt-1	2.82	**	(2.53,3.16)	2.67	**	(2.39,2.99)	2.88	**	(2.51,3.29)
LWH(WH ≥ 60)	1.23	**	(1.12,1.36)	1.11	†	(0.98,1.25)	1.16	*	(1.03,1.30)
Covariates	Yes			Yes			Yes		
*N*	11,010			11,010			7060		
(2) Change to use a set of dummy variables of LWH									
MIt-1	2.82	**	(2.52,3.16)	2.66	**	(2.38,2.97)	2.88	**	(2.52,3.30)
WH (Ref. WH<35 h)	1			1			1		
WH35-39	0.88	†	(0.76,1.01)	0.79	**	(0.66,0.95)	1.02		(0.86,1.21)
WH40-49	0.97		(0.85,1.11)	0.90		(0.76,1.06)	1.09		(0.93,1.27)
WH50-59	1.08		(0.95,1.23)	0.99		(0.85,1.15)	1.11		(0.95,1.28)
WH ≥ 60	1.22	**	(1.08,1.37)	1.03	†	(0.87,1.17)	1.23	**	(1.07,1.42)
Covariates	Yes			Yes			Yes		
*N*	11,010			11,010			7060		
(3) Change the definition of MI (TMH ≥ 12)									
MIt-1	2.65	**	(2.29,3.08)	2.62	**	(2.32,2.96)	2.98	**	(2.43,3.65)
LWH(WH ≥ 50)	1.20	**	(1.10,1.31)	1.12	*	(1.02,1.26)	1.13	*	(1.02,1.25)
Covariates	Yes			Yes			Yes		
*N*	11,010			11,010			7060		
(4) Change the definition of MI (TMH ≥ 16)									
MIt-1	1.77	**	(1.42,2.20)	2.14	**	(1.82,2.52)	1.73	**	(1.42,2.41)
LWH(WH ≥ 50)	1.21	**	(1.11,1.32)	1.14	*	(0.98,1.25)	1.13	*	(1.02,1.25)
Covariates	Yes			Yes			Yes		
*N*	11,010			11,010			7060		
(5) Change the definition of MI (TMH ≥ 20)									
MIt-1	1.37	**	(1.07,1.77)	1.75	**	(1.42,2.16)	1.45	*	(1.01,2.07)
LWH (WH ≥ 50)	1.21	**	(1.11,1.32)	1.15	*	(1.03,1.28)	1.13	*	(1.02,1.25)
Covariates	Yes			Yes			Yes		
*N*	11,010			11,010			7060		
(6) Using the samples excluding farmers									
MIt-1	2.91	**	(2.50,3.38)	2.76	**	(2.36,3.23)	3.01	**	(2.48,3.66)
LWH(WH ≥ 50)	1.39	**	(1.20,1.61)	1.17	†	(0.98,1.39)	1.13	†	(0.96,1.34)
Covariates	Yes			Yes			Yes		
*N*	6681			6681			3996		
(7) Using the logit regression model									
MIt-1	2.74	**	(2.29,3.28)	2.62	**	(2.18,3.13)	3.04	**	(2.42,3.82)
LWH(WH ≥ 50)	0.20	**	(0.11,0.29)	0.12	*	(0.01,0.22)	0.11	*	(0.01,0.22)
Covariates	Yes			Yes			Yes		
*N*	11,010			11,010			7060		

Note: We used the dynamic lagged variable logistic models in the estimations (1)–(6) and presented the results of the odds ratio; we used the dynamic lagged variable logit regression model in the estimation (7) and presented the results of the coefficients. Logit: logistic regression model. LVt-1: model using the indicator for the risk of mental illness in the previous period. LVt-2: model using the indicator for the risk of mental illness in the past two periods. OR: odds ratio. CI: confidence interval. MI: an indicator for the risk of mental health. TMH: total score for the risk of mental health. MIt-1: the indicator of the risk of mental illness in the prior survey year. LWH: an indicator variable for long working hours. LWH (≥50) and LWH (≥60) are dummy variables equal to 1 when an individual answered that their weekly work hours were 50 or 60 h. or more, respectively, and 0 otherwise. WH: weekly working hours. The covariates including years of education, years of working experience and its squared, female, ethnicity (Han dummy), party membership, urban, married, number of family members, occupation (manager and technician, staff and service, and other occupations), industry sector (manufacture dummy), participation in public pension, participation in medical insurance, region (central, west) and year dummy variables (2014, 2016) were controlled. The reference group is operation workers, east region, and 2018 year. ** *p* < 0.01, * *p* < 0.05, † *p* < 0.10.

**Table 4 ijerph-20-01641-t004:** Estimated associations between long working hours and MI by sex.

	(1) Men			(2) Women		
	OR.		95% CI	OR.		95% CI
MIt-1	2.84	**	(2.41,3.35)	2.53	**	(2.17,2.95)
LWH(WH ≥ 50)	1.11		(0.95,1.30)	1.13	†	(0.98,1.31)
Covariates	Yes			Yes		
*N*	5756			5254		
Log likelihood	−2148.93			−2394.02		
Pseudo R2	0.11			0.10		

Note: Obtained results from the dynamic logistic models with lagged explanatory variables (dynamic LVt-1 model). The covariates including years of education, years of working experience and its squared, female, ethnicity (Han dummy), party membership, urban, married, number of family members, occupation (manager and technician, staff and service, and other occupations), industry sector (manufacture dummy), participation in public pension, participation in medical insurance, region (central, west) and year dummy variables (2014, 2016) were controlled. ** *p* < 0.01, † *p* < 0.10.

**Table 5 ijerph-20-01641-t005:** Estimated associations between long work hours and MI by occupational group.

	(1) White Collar			(2) Pink Collar			(3) Blue Collar		
	OR.		95% CI	OR.		95% CI	OR.		95% CI
MIt-1	2.66	**	(1.66,4.26)	2.34	**	(1.80,3.04)	3.22	**	(2.53,4.10)
LWH(WH ≥ 50)	1.65	*	(1.05,2.59)	1.04		(0.82,1.33)	1.09		(0.87,1.36)
Covariates	Yes			Yes			Yes		
*N*	921			2287			2754		
Log-likelihood	−297.83			−890.01			−1101.25		
Pseudo R2	0.14			0.10			0.16		

Note: Obtained results from the dynamic logistic models with lagged explanatory variables (dynamic LVt-1 model). The covariates including years of education, years of working experience and its squared, female, ethnicity (Han dummy), party membership, urban, married, number of family members, occupation (manager and technician, staff and service, and other occupations), industry sector (manufacture dummy), participation in public pension, participation in medical insurance, region (central, west) and year dummy variables (2014, 2016) were controlled. ** *p* < 0.01, * *p* < 0.05.

**Table 6 ijerph-20-01641-t006:** Estimated associations between long work hours and MI by employment sector group.

	(1) Self-Employed Individual			(2) Employee in Micro-Firm			(3) Employee in Large Firm		
	OR.		95% CI	OR.		95% CI	OR.		95% CI
MIt-1	2.66	**	(1.89,3.72)	3.18	**	(2.37,4.26)	2.14	**	(1.38,3.31)
LWH(WH ≥ 50)	1.03		(0.75,1.43)	1.29	*	(0.97,1.70)	1.02		(0.67,1.57)
Covariates	Yes			Yes			Yes		
*N*	1438			1567			978		
Log likelihood	−551.00			−672.91			−329.78		
Pseudo R2	0.13			0.13			0.10		

Note: Obtained results from the dynamic logistic models with lagged explanatory variables (dynamic LVt-1 model). The covariates including years of education, years of working experience and its squared, female, ethnicity (Han dummy), party membership, urban, married, number of family members, occupation (manager and technician, staff and service, and other occupations), industry sector (manufacture dummy), participation in public pension, participation in medical insurance, region (central, west) and year dummy variables (2014, 2016) were controlled. ** *p* < 0.01, * *p* < 0.05.

## Data Availability

Raw data of CFPS are available from the following URLs: (http://opendata.pku.edu.cn/en) (accessed on 2 March 2020).

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
