# Peer review of "Impact of Long Working Hours on Mental Health: Evidence from China"

_ijerph, 2023, doi:10.3390/ijerph20021641_

Round 1
Reviewer 1 Report
Dear authors
I find your study very pertinent and of great interest to the scientific community. Although there are many studies that intersect long working hours with mental health, It's never enough to insist on this theme until it brings about a transformation in business conduct and labor laws. Psychosocial hazards are factors in the design or management of work that increase the risk of work related stress and can lead to psychological or physical harm. These may include work organization, social factors such as excessive working hours, poor leadership or bullying, as well as physical aspects of the work environment equipment and exposure to several products. They can b present in all organizations, across all sectors and countries. Literature points stress, depression, and suicidal thoughts by multivariate logistic regression analysis based on working hours, which was adjusted for sex, age, marriage status, region, and educational level.
Also suggests that longer working hours are associated with poorer mental health status and increasing levels of anxiety and depression symptoms with a positive correlation between these symptoms and sleep disturbances.
Studies say that the the average working hours of the workforce exceed the requirements of the Labor Law of China, which thus makes it necessary to limit the excessive working hours of the workforce and promote the regularization of working hours in the labor market.
These are some of the questions that I would like you to be able to develop in your theoretical framework and in the conclusions. Despite these small notes, the methodology seems to me to be very good and all the presentation of the results. I would like to highlight your ability to address a topic that certainly is still little talked about in China and that can certainly bring to debate an important issue for the welfare of the population.
Best wishes
Author Response
Thank you very much for giving me so many useful suggestions and comments, which helped me to improve the quality of the manuscript. I have considered each comment carefully and corrected the manuscript as follows:
- I have adjusted the contents of Introduction part to emphasize the theory descriptions at lines 41-57.
- According to your suggestions, I have added the significances of this study in Conclusion part at lines 385-387 as follows:
…We expect the Chinese experience to provide valuable lessons for other countries looking to improve their nations’ mental health and population’s welfare.
Reviewer 2 Report
1. The introduction and theoretical analysis are too simple to fully reflect the inherent logical relationship between mental health and long working hours as influencing factors.
2. Why we're working with individuals in the urban and rural areas in China selected as cases? Does the mental health problem in the case area have any salient characteristics? Is the problem representative? These issues require clearer clarification in the introduction.
3. Manuscripts lack an explanation of statistical details, as described above. Despite the volume of data, there is no specific information about the different variables used in this study. Why are these variables important?
4. A more in-depth analysis of the mechanisms by which long working hours affect mental health is needed in terms of the theoretical implications for the theoretical value of this study.
5. A more in-depth and targeted analysis of practical implications is needed to provide a better reference or guide for the case area.
6. Some limitations in the study area analysis, the logical theoretical relationship of long working hours and mental health, the basis of the selected plots and samples, and targeted suggestions should be more clearly illustrated.
Author Response
[Reply] Thank you very much for giving me so many useful suggestions and comments, which helped me to improve the quality of the manuscript. I have considered each comment carefully and corrected the manuscript as follows.
- The introduction and theoretical analysis are too simple to fully reflect the inherent logical relationship between mental health and long working hours as influencing factors.
[Reply] Thank you very much for the helpful comments. I have tried to explain the association between long work time and mental health from the occupational health and human resource management and labor and family economics theory perspectives. I have adjusted the contents of Introduction part to emphasize the theory descriptions at lines 41-57.
- Why we're working with individuals in the urban and rural areas in China selected as cases? Does the mental health problem in the case area have any salient characteristics? Is the problem representative? These issues require clearer clarification in the introduction.
[Reply] Thank you very much for the helpful comments. According to your suggestions, I have added the contents to explain why I selected China as an analyzed case in Introduction Part as follows (at lines29-40):
This study focuses on the relationship between long working hours and MI in China. China was selected as an analyzed target because it is the most populated country in the world and has the highest number of patients with MI, globally. According to the World Health Organization, approximately 54 and 41 million people in China suffer from depression and anxiety disorders, respectively; the proportion of Chinese people with MI is more than 12% of that worldwide [3]. Hence, a public health policy aimed at improving Chinese mental health status may contribute to the population welfare worldwide. In addition, regarding overtime work in China, although the Labor Law of the People’s Republic of China regulates the standard working hours (around 40 hours weekly), work hours are actually longer in China’s privately owned enterprises and the informal sector (e.g., the self-employed) [4, 5]. Although long working hours likely affect a worker’s mental health in China, empirical studies on the issue that focus specifically on China are scarce. This study attempts to fill the above-mentioned gaps.
- Manuscripts lack an explanation of statistical details, as described above. Despite the volume of data, there is no specific information about the different variables used in this study. Why are these variables important?
[Reply] Thank you very much for the helpful comments. According to your suggestions, I added the contents to describe the confounder variables as follows (at lines 186-203).
- A more in-depth analysis of the mechanisms by which long working hours affect mental health is needed in terms of the theoretical implications for the theoretical value of this study.
[Reply] Thank you very much for the helpful comments. I very agreed with your suggestions. I have added the contents as follows (at lines 369-373).
…Fifth, because we could not obtain appropriate information (e.g., the job allocation, effort or willingness to work, work environment in workplace) from the questionnaire items of the CFPS, an empirical study to explore the mechanism of the effect of long working hours on mental health status should be explored in the future.
- A more in-depth and targeted analysis of practical implications is needed to provide a better reference or guide for the case area.
[Reply] Thank you very much for the helpful comments. I have done various analyses such as comparing the differences between education, age and urban and rural groups, I found the differences between these group are not statistically significant. Thus, I have not presented these results. I only presented the results that are statistically significant in the manuscript.
- Some limitations in the study area analysis, the logical theoretical relationship of long working hours and mental health, the basis of the selected plots and samples, and targeted suggestions should be more clearly illustrated.
[Reply] Thank you very much for the helpful comments. According to your suggestions, I added the contents in limitation part as follows (at lines361-379):
This study has several limitations. …Second, although the sample for the baseline of the CFPS was drawn through multistage probability with implicit stratification, it may also maintain the plots or sample selection bias in the survey. Third, although we used dynamic models with lagged long working hours variables, we could not identify the underlying causality of long working hours affecting mental health, which should be investigated in greater depth. Fourth, as China is a developing and emerging economy country, and the influence of government (or Communist Party of China organization) on firms is greater that in developed countries [46, 47], it is important to account for international comparisons between China and other counties while considering these institutional differences…
Reviewer 3 Report
In the reviewed article, the authors described the research in a correct way. However, reading the text, I ask myself why these studies. There is a lack of assumptions and research context. It is well known that long working hours have a negative impact on employee productivity. And here research was carried out, the results obtained were analyzed, but there is no practical or scientific effect. The research was conducted in 2014-2018, i.e. before the pandemic. And now we have completely different working conditions. Lessons learned cannot be adapted to the current situation.
In conclusion, unfortunately, the data are outdated, which results in a negative assessment of the text.
Author Response
[Reply] Thank you very much for giving me so many useful suggestions and comments, which helped me to improve the quality of the manuscript. I have considered each comment carefully and corrected the manuscript as follows.
In the reviewed article, the authors described the research in a correct way. However, reading the text, I ask myself why these studies. There is a lack of assumptions and research context. It is well known that long working hours have a negative impact on employee productivity. And here research was carried out, the results obtained were analyzed, but there is no practical or scientific effect. The research was conducted in 2014-2018, i.e. before the pandemic. And now we have completely different working conditions. Lessons learned cannot be adapted to the current situation.
In conclusion, unfortunately, the data are outdated, which results in a negative assessment of the text.
[Reply] Thank you very much for the helpful comments.
- This study attempts to explore the association between long working hours and mental health illness for China based on a national longitudinal survey to address the reverse causality and heterogeneity issues that has been not considered in the previous studies.
- I agreed with your arguments on the practical effect during the COVIN-19 pandemic period when the employment status has changed greatly worldwide. The COVIN-19 pandemic shock may increase the risk of becoming mental illness significantly. Because I could not obtain the survey data during the COVIN-19 pandemic shock, I could not analyze the issue including the COVIN-19 pandemic period. After considering the comments carefully, I added the contents on the COVIN-19 pandemic in Limitation part (at lines369-376) as follows:
…Fifth, because we could not obtain appropriate information (e.g., the job allocation, effort or willingness to work, work environment in workplace) from the questionnaire items of the CFPS, an empirical study to explore the mechanism of the effect of long working hours on mental health status should be explored in the future. Finally, the COVID-19 pandemic has immensely changed the employment status [48, 49], thus damaging individuals’ mental health status worldwide [50, 51]. Further study on employee mental health in light of the COVID-19 pandemic is a new challenge.
Reviewer 4 Report
Thank you very much for giving me the opportunity to review the manuscript entitled “Impact of long working hours on mental health: Evidence from China”. I have read it with great interest. Using dynamic regression models with lagged long working hours variables, this study investigated the association between long working hours and the risk of mental illness nationwide. Data came from the China Family Panel Studies conducted in 2014, 2016, and 2018. Long working hours were found to be positively and significantly associated with the increased risk of mental illness. Subgroups analyses found that the association was significant only in women, only in white-collar workers, and only in informal sector workers. It was suggested that the results would provide rich evidence of effects of long working hours on mental health in China.
The introduction section is well written. It provides a good summary of previous studies, adequately justifies the present study, and successfully clarify the meaning of knowing the current results. The method section adequately describes the research procedures. However, I feel that the result and discussion benefit from revisions as follows.
<1> It is difficult to understand results shown in Table 1. Especially, I can not understand what values in the table mean except Weekly WH, Education, and Age. For example, the value for the (a) Total raw of Women line is 0.49. I can not understand what this 0.49 means.
<2> Table 3, (3) Limit samples to only employees, “(1.20, 1.61)”. If my reading is correct, (1.20, 1.60) may be correct.
<3> Line 224-226, “the positive effect of long working hours on the risk of MI is more significant for women than for men (p < 0.1 for women)”. I feel that the term “significant” is not correct. The sample size is very large, and I do not think that p value of <.10 is meaningful. I would write “effect of long working hours on the risk of MI is not significant for both men and women”.
<4> Line 227-229, “the positive effect of long working hours on the risk of MI is significantly higher for the white-collar group (OR: 1.65, p < 0.05) than for the pink- and blue-collar groups”. I would write “the positive effect of long working hours on the risk of MI is significant only the white-collar group (OR: 1.65, p < 0.05)”.
<5> Line 268, “The differences may be attributable to the different data and models: we used the three-wave national longitudinal survey data from 2014 to 2018 and the dynamic lagged variable model to address the initial dependent and reverse causality issues”. If my reading is correct, author interpreted that weaker associations of long working hours on the risk of MI would be attributable to statistical control of the initial value (MIt-1). However, results for the initial value are not shown in the current manuscript. If the coefficients for the initial values are shown and certainly significant, this interpretation would be more persuasive.
Author Response
[Reply] Thank you very much for giving me so many useful suggestions and comments, which helped me to improve the quality of the manuscript. I have considered each comment carefully and corrected the manuscript as follows.
Thank you very much for giving me the opportunity to review the manuscript entitled “Impact of long working hours on mental health: Evidence from China”. I have read it with great interest. Using dynamic regression models with lagged long working hours variables, this study investigated the association between long working hours and the risk of mental illness nationwide. Data came from the China Family Panel Studies conducted in 2014, 2016, and 2018. Long working hours were found to be positively and significantly associated with the increased risk of mental illness. Subgroups analyses found that the association was significant only in women, only in white-collar workers, and only in informal sector workers. It was suggested that the results would provide rich evidence of effects of long working hours on mental health in China.
The introduction section is well written. It provides a good summary of previous studies, adequately justifies the present study, and successfully clarify the meaning of knowing the current results. The method section adequately describes the research procedures. However, I feel that the result and discussion benefit from revisions as follows.
<1> It is difficult to understand results shown in Table 1. Especially, I can not understand what values in the table mean except Weekly WH, Education, and Age. For example, the value for the (a) Total raw of Women line is 0.49. I can not understand what this 0.49 means.
[Reply] Thank you very much for the helpful comments. After considering the comments carefully, I corrected the manuscript as follows:
- I changed the format of Table1 to make the results to be understanded easily. I mainly changed a part of results from figure to percent (%).
- I added the detailed descriptions on the results as follows (at lines193-199):
Second, the demographic, family, and workplace factors differ in the long- and standard or less working hours groups. For example, the proportion of women was 49% for the total sample, 42% for long working hours group, and 55% for standard or less working hours group, suggesting a gender distribution gap in two groups. The proportion of operation workers was 26% for the total sample, 35% for long working hours group, and 19% for standard or short working hours group, indicating the occupational distribution differs among different working hour groups.
<2> Table 3, (3) Limit samples to only employees, “(1.20, 1.61)”. If my reading is correct, (1.20, 1.60) may be correct.
[Reply] Thank you very much for the helpful comments. I calculated the 95% CI again and found they are right. It was caused by the rounding up calculations. The original figure are (1.195899, 1.605814).
<3> Line 224-226, “the positive effect of long working hours on the risk of MI is more significant for women than for men (p < 0.1 for women)”. I feel that the term “significant” is not correct. The sample size is very large, and I do not think that p value of <.10 is meaningful. I would write “effect of long working hours on the risk of MI is not significant for both men and women”.
[Reply] Thank you very much for the helpful comments. According to your suggestions, I corrected the manuscript as follows (at lines248-251):
First, although the coefficients of long working hours are positive values for both men and women, they are not statistically significant for either. The positive effect of long working hours on the risk of MI is slight greater for women than for men (1.13, p < 0.1 for women).
<4> Line 227-229, “the positive effect of long working hours on the risk of MI is significantly higher for the white-collar group (OR: 1.65, p < 0.05) than for the pink- and blue-collar groups”. I would write “the positive effect of long working hours on the risk of MI is significant only the white-collar group (OR: 1.65, p < 0.05)”.
[Reply] Thank you very much for the helpful comments. According to your suggestions, I corrected the manuscript as follows (at lines252-253):
Second, the positive effect of long working hours on the risk of MI is significant only the white-collar group (OR: 1.65, p < 0.05).
<5> Line 268, “The differences may be attributable to the different data and models: we used the three-wave national longitudinal survey data from 2014 to 2018 and the dynamic lagged variable model to address the initial dependent and reverse causality issues”. If my reading is correct, author interpreted that weaker associations of long working hours on the risk of MI would be attributable to statistical control of the initial value (MIt-1). However, results for the initial value are not shown in the current manuscript. If the coefficients for the initial values are shown and certainly significant, this interpretation would be more persuasive.
[Reply] Thank you very much for the helpful comments. According to your suggestions, I corrected the manuscript as follows:
- I added the estimation results of the initial MI variable in Tables 2-6.
- I added the contents to describe the results about the initial value at lines to emphasize that use of the initial dependent variable is appropriate as follows (at lines 212-214).
…Furthermore, the coefficients of the initial independent variables (MIt-1) are significant at 1% levels, suggesting an initial problem. Hence, using the dynamic model is valid.
Round 2
Reviewer 2 Report
The authors have made revisions as directed. In addition, it is recommended that the authors explain research limitations using quantitative studies and expectations for future studies in this paper.
Author Response
[Reply] Thank you very much for the helpful comments. I have added the contents to explain research limitations using quantitative studies and pointed out what we should do in the future research at lines 902-903, 905-906, 909-910, 911-913, 917-918, 920-923, and 926 in the revised (R2) manuscript.
This study has several limitations. First, the results may be subject to measurement [49, 50]. For example, the women may report a higher risk of MI than men, and the less educated and the older groups might not report the correct status of their mental health because of the lack of literacy, so future studies should aim to adopt objective indicators of MI. The measurement of long working hours (e.g., measurement scale, cut-off point, and the definition of long working hours) differs across empirical studies [28], and an international comparison cannot be easily performed. Thus, international comparison research based on similar definitions should be pursued in the future. Second, although the sample for the baseline of the CFPS was drawn through multistage probability with implicit stratification, it may also maintain the plots or sample selection bias in the survey. Third, although we used dynamic models with lagged long working hours variables to address the reverse causality problem, we could not identify the underlying causality of long working hours affecting mental health such as individual heterogeneity and the self-selection problem, which should be investigated in greater depth using various quantitative study models (e.g., difference-in-differences, instrumental variable methods). Fourth, as China is a developing and emerging economy country, and the influence of government (or Communist Party of China organization) on firms is greater that in developed countries [51, 52], it is important to account for international comparisons between China and other counties while considering these institutional differences. We should control these institutional and cultural factors in any international comparison. Fifth, because we could not obtain appropriate information (e.g., the job allocation, effort or willingness to work, work environment in workplace) from the questionnaire items of the CFPS, future works could conduct alternative surveys such as employer-employee surveys that include these workplace items and employ empirical methods to explore the mechanism by which long working hours affects mental health status.

Reviewer 3 Report
I still stand by my earlier comments. The authors in the corrected version added only a few lines of text and it does not affect my assessment of the article. The problem here is not the text, but the research and its scientific significance are still a problem.
Author Response
[Reply] Thank you very much for giving me so many useful suggestions and comments, which helped me to improve the quality of the manuscript. I have considered each comment carefully and corrected the manuscript as follows.
- In the reviewed article, the authors described the research in a correct way. However, reading the text, I ask myself why these studies. There is a lack of assumptions and research context. It is well known that long working hours have a negative impact on employee productivity. And here research was carried out, the results obtained were analyzed, but there is no practical or scientific effect. The research was conducted in 2014-2018, i.e. before the pandemic. And now we have completely different working conditions. Lessons learned cannot be adapted to the current situation. In conclusion, unfortunately, the data are outdated, which results in a negative assessment of the text. I still stand by my earlier comments. The authors in the corrected version added only a few lines of text and it does not affect my assessment of the article. The problem here is not the text, but the research and its scientific significance are still a problem.
[Reply] Thank you very much for the helpful comments. I have considered these comments and tried to correct the manuscript according to your suggestions as follows.
- I have added the contents to explain the significance of this study in the Introduction part (at lines 27-36) as follows; I have added the related literatures to clarify the significance of this paper among the existing literatures. I have added these related literatures in the reference list.
…As medical care expenses for MI are high [1,3] and MI may reduce labor productivity, negatively affecting human capital accumulation in most countries, exploring the determinants of MI is a critical issue in the public health field.
Some empirical studies revealed that individual attributes (e.g., education, age, sex) [4-7], social capital [8, 9], and life events (e.g., marriage, fertility) [10-12] may affect the risk of developing MI. Furthermore, the work-life conflict may increase the risk of MI [13,14]. Among the various work environment factors that may harm workers’ mental health status, prior empirical works identify long working hours as a risk factor [4-7, 15-23]. However, most studies concentrated only on developed countries, and evidence from China is scarce.
- I have added the assumptions in the introduction part at lines 106-109 as follows:
According to the job demand-control [26] and effort-reward imbalance models [27], and the previous studies above, we hypothesize that long working hours may negatively affect Chinese workers’ mental health status, and the negative effect will differ by group. We perform an empirical study to prove our hypotheses in the following.
- Thank you very much for your suggestion on the resent significance of this study. I have tried to link the policy implications considering the COVID-19 pandemic in the past three years. I have added the contents to state the resent policy meaning of this study in conclusion part at lines927-1121 as follows:
…Although we analyzed data from a survey conducted before the COVID-19 pandemic, employment has slowly returned to normal in the recent period in most countries. The Chinese government began winding down its policy of closing down cities in December 2022. Because the COVID-19 pandemic negatively affected economic growth worldwide, which led to dramatic decreases in labor demand in the past three years, employers will extend employees’ working hours to reduce labor costs. Hence, the problem of long working hours may become much more serious in the future. The results from this empirical study and the existing research conducted in developed countries also suggest that the government should enforce working hour regulations to reduce the risk of MI during or after the COVID-19 pandemic period.

Reviewer 4 Report
All concerns have been addressed. Thank you very much for thorough reply to my comments.
Author Response
Thank you very much for givimg me the second round comments. I am very pleasure that you have agree with these corrections. Thank you very much again for your helpful comments and suggestions.